# LSW-Net: A Learning Scattering Wavelet Network for Brain Tumor and Retinal Image Segmentation

**Ruihua Liu [1],\*, Haoyu Nan [1], Yangyang Zou [1], Ting Xie [2] and Zhiyong Ye [2]**

[1] School of Artificial Intelligence, Chongqing University of Technology, Chongqing 401135, China
[2] College of Science, Chongqing University of Technology, Chongqing 400054, China
\* Correspondence: lruih@cqut.edu.cn

**Abstract:** Convolutional network models have been widely used in image segmentation. However, there are many types of boundary contour features in medical images which seriously affect the stability and accuracy of image segmentation models, such as the ambiguity of tumors, the variability of lesions, and the weak boundaries of fine blood vessels. In this paper, in order to solve these problems we first introduce the dual-tree complex wavelet scattering transform module, and then innovatively propose a learning scattering wavelet network model. In addition, a new improved active contour loss function is further constructed to deal with complex segmentation. Finally, the equilibrium coefficient of our model is discussed. Experiments on the BraTS2020 dataset show that the LSW-Net model has improved the Dice coefficient, accuracy, and sensitivity of the classic FCN, SegNet, and At-Unet models by at least 3.51%, 2.11%, and 0.46%, respectively. In addition, the LSW-Net model still has an advantage in the average measure of Dice coefficients compared with some advanced segmentation models. Experiments on the DRIVE dataset prove that our model outperforms the other 14 algorithms in both Dice coefficient and specificity measures. In particular, the sensitivity of our model provides a 3.39% improvement when compared with the Unet model, and the model's effect is obvious.

**Keywords:** image segmentation; wavelet scattering; loss function; active contour; medical image

## 1. Introduction

Image segmentation is a class of image processing problems, and its task is to divide an image into two or more meaningful regions. The accuracy of image segmentation is particularly important in practical applications. In particular, biomedical image segmentation is prominent in clinical analysis, diagnosis, treatment planning, and the measurement of disease progression. Traditional image segmentation methods, such as the threshold method [1], region growing method [2], level set method [3–5], etc., have struggled to meet the need for accurate image segmentation in the context of big data.

In recent years, deep neural networks have made great progress in various artificial intelligence tasks including image recognition and image segmentation. A convolutional neural network (CNN) [6] introduces semantic information when segmenting objects; thereby, injecting new vitality into semantic segmentation research. Fully convolutional network models [7–9] based on CNN architecture have achieved excellent performance in automatic medical image segmentation, which further promotes the application of deep learning in image segmentation for applications such as brain tumor segmentation [10]. SegNet [11] adopts the encoder–decoder structure and transfers the pixel index value of the maximum pooling operation in the encoding process into the decoder, which not only retains the detailed information of the pixels but also improves the accuracy of semantic segmentation. The Attention Unet network (At-Unet) [12] adds an attention gating unit to the Unet model to provide pixel level attention for the feature map. The network tends to focus on feature points with more information and improves the feature extraction ability

of the model. The Deeplab network [13,14] obtains multiscale context information by cascading atrous convolutions with different atrous rates, and then introduces a conditional random field to enhance the relevance of contextual semantic information, which in turn improves the segmentation accuracy. Although the above-mentioned network models have improved the image segmentation accuracy of some datasets, they are still unable to accurately extract the boundary features of brain tumors in images. This is due to the invasiveness of the imaging process and the ambiguity between biological forms, as is the case between tumors and adjacent organs or changes in lesions over different periods. This invasiveness and ambiguity can lead to the discontinuity of some segmentation boundaries, as shown in Figure 1a. When comparing Figure 1a,b, there are many discontinuous segmentations in Figure 1a.

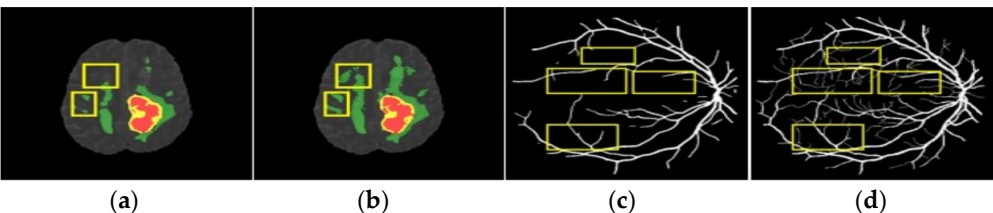

**Figure 1.** Medical image segmentation. (**a**) At-Unet; (**b**) Brain tumor ground truth; (**c**) AC-Loss; (**d**) Retinal ground truth, where the yellow boxes indicate the regions where the methods are missegmented compared to ground truth.

At the same time, many researchers are devoted to establishing the minimal loss energy function model. A level set function is proposed, that is, the region term of the CV energy function is used as the loss function, from which the CNN model can learn the spatial information of the image, which can improve the accuracy of image segmentation [14]. In order to solve the boundary error segmentation problem, an active contour loss function (AC-Loss) is constructed [15]. The AC-Loss function fully considers the internal and external areas of the segmented object, and the perimeter of the boundary. Unfortunately, some experiments have shown that when dealing with biomedical images with complex boundaries, such as retinal vessel images, because the AC-Loss function constrains the perimeter of the segmentation object boundary, it also limits the model's ability to segment small boundaries. The under-segmentation phenomenon of fine blood vessels is avoided in Figure 1c. By comparing Figure 1c,d, there are many small blood vessels that can be seen in Figure 1d that are not segmented in Figure 1c.

The problem of complex boundary contour features in medical images, also increases the difficulty of image boundary feature extraction and characterization in deep neural network learning. Inspired by the dual-tree wavelet scattering transform, we propose a boundary feature extraction module which can improve the network's ability to extract image boundary features. Specifically, the process can be described as follows: First, the dual-tree complex wavelet scattering transform is used to separate the high-frequency and low-frequency features of the feature map. Second, a convolution operator is adopted to extract low-frequency body features and high-frequency boundary features. Finally, the dual-tree complex wavelet scattering transform is then built into a fully convolutional network model, and a new learning scattering wavelet network (LSW-Net) semantic segmentation model is designed through end-to-end data-driven scattering learning transform features. In order to enhance its ability to extract image boundary contour information, the network utilizes the Unet network [8] as the backbone network and introduces the dual-tree complex wavelet scattering transform (DTCWT-Scat) during downsampling for boundary feature extraction. In order to further improve the network model's ability to extract complex boundary contours, an improved active contour loss function (IAC-Loss) is further constructed on the basis of the LSW-Net network. This loss function not only improves the network's sensitivity to small boundaries, it also better solves the problem of the under-segmentation of boundary contours.

Our main contributions are summarized as follows:

- In order to separate the high-frequency and low-frequency features of the feature map during downsampling, we introduce the DTCWT-Scat module into the Unet and innovatively propose the LSW-Net model.
- We design an improved active contour loss function, which can improve sensitivity to small boundaries and can better solve the problem of boundary under-segmentation.
- Through BraTS brain tumor segmentation experiments, our LSW-Net network has advantages when compared with traditional FCN, SegNet, At-Unet, and some advanced segmentation algorithms in terms of Dice coefficient, accuracy, sensitivity, and other indicators.
- Through the DRIVE retinal vessel segmentation experiments, the effectiveness and robustness of the LSW-Net + IAC-Loss model are illustrated.

## 2. Related Work

### 2.1. Dual-Tree Complex Scattering Wavelet Transform

Wavelet transform is a local waveform transform that can provide local representation of multiscale signals in both time and frequency domains. S. Mallat first proposed a wavelet scattering network with a non-feedback structure [16]. This network can not only present the image energy distribution in the frequency domain, but also maintain stability against small deformations. This partially makes up for the shortcomings of the CNN model, including small object segmentation and image boundary extraction capabilities. Some scholars have also actively tried to combine the wavelet algorithm with the CNN model. Oyallon [17] used a wavelet scattering network to replace the first layer of a residual network. The modified residual network produces roughly the same performance as the original residual network, but the training parameters are greatly reduced. Rodriguez [18] proposed a deep adaptive wavelet network to capture basic information from the input data for image classification. Through experiments on three image classification datasets, it was found that the model achieved high accuracy and also reduced training parameters. Recently, Cotter [19] proposed a dual-tree complex wavelet scattering network. After being combined with a CNN model, it achieves high accuracy in image classification tasks as well as fast inference ability. Figure 2 shows the output results of the first-order dual-tree complex wavelet scattering of brain tumor MRI images, including one low-frequency signal and high-frequency signals in six directions. The low-frequency signal is the main feature of the image, and the six high-frequency signals are the boundary feature of the image.

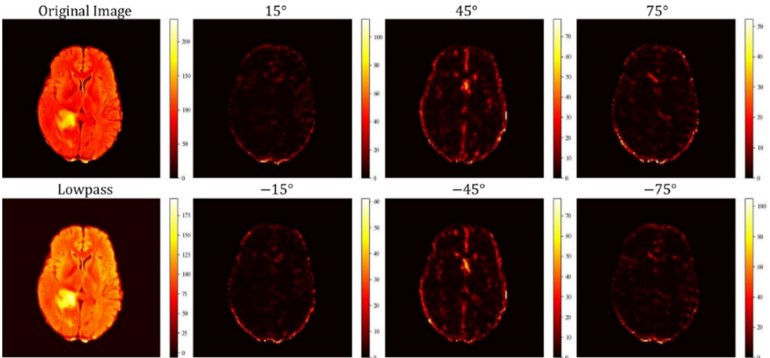

**Figure 2.** Visualization of the first-order dual-tree complex wavelet scattering transform of brain tumor MRI images.

### 2.2. Related Loss Functions

While the widely used cross-entropy loss function (CE-Loss) is not sensitive to the segmentation of small object boundaries, when the existing model is trained, the network model will optimize its parameters using a gradient descent method according to the loss function error. Figure 3 shows that the CE-Loss function does not perform very well for

cases with small boundaries or a small number of misclassified boundaries. To solve this problem, Williams [15] et al. proposed the AC-Loss loss energy function, which can be described as; where the Region item is the area of the segmentation region, the AC item is the boundary length of the segmentation object, the item is the area of the segmentation region, and the item is the boundary length of the segmentation object. In order to reduce false boundary segmentation, this energy function is expected to minimize the area energy of the segmentation region and the energy of the segmentation target boundary length during model training.

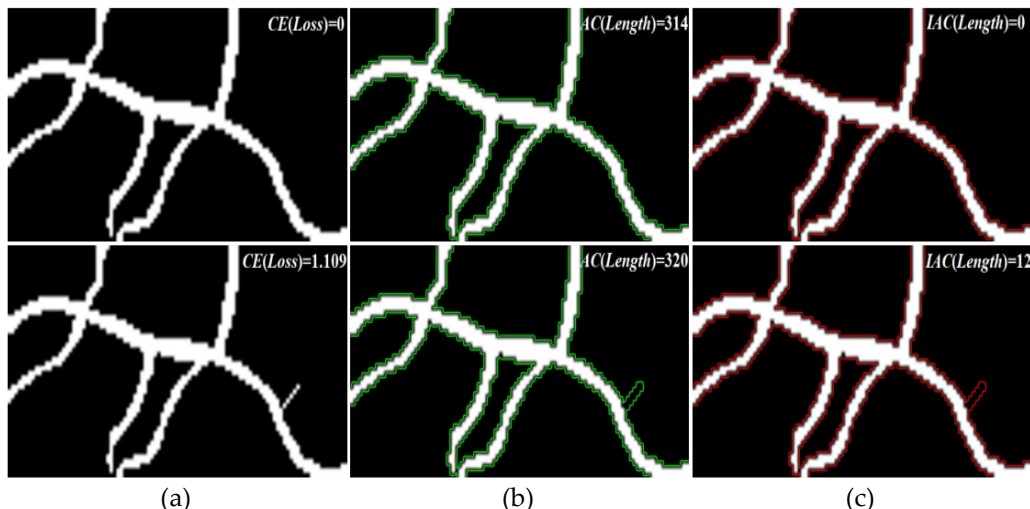

**Figure 3.** Comparison example of boundary contour segmentation. (**a**) CE-Loss; (**b**) AC-Loss; (**c**) IAC-Loss.

Unfortunately, it can be observed that the AC-Loss function is more sensitive than the CE-Loss term when there is a small boundary in complex medical images, such as the retinal vessel shown in Figure 3. However, when the target segmentation is completely correct, the AC-Loss term still maintains a high error value. As the model is further trained, this will further reduce the length of the segmentation target boundary resulting in under-segmentation for some boundaries.

## 3. Proposed Method

In this section, we first construct the DTCWT-Scat module and then propose a novel LSW-Net network model after introducing the DTCWT-Scat module into the Unet network. Furthermore, in order to solve the small target segmentation task, a new IAC-Loss function is designed. Finally, we document the LSW-Net algorithm and the IAC-Loss function calculation algorithm.

### 3.1. Learning Scattering Wavelet Network

Wavelet scattering can extract image texture features and boundary information but cannot make full use of contextual semantic information for image segmentation. FCN integrates multiscale contextual information through multilayer pooling and subsampling; however, it is still unable to distinguish boundary information from overall information. The natural solution of combining the two functional modules can not only enhance the complementarity between the boundary information and the global information but also improve the classification accuracy of image boundaries. Therefore, we designed a novel LSW-Net model that combines a wavelet scattering network and a fully convolutional network, which is based on the encoder–decoder structure of the fully convolutional network [20]. The LSW-Net framework can be described in detail as follows: First, the dual-tree complex wavelet scattering transform [19] is added during downsampling in order to effectively separate the high-frequency features and low-frequency features of the

feature map. Second, the convolution operator is used to select the low-frequency main features and high-frequency boundary features of the feature map, respectively. Finally, we concentrate these features. The decoder is a process that uses a multilayer upsampling method to gradually restore its original resolution. The algorithm is shown in Algorithm 1.

---

**Algorithm 1:** Learning Scattering Wavelet Network

---

**Input:** Preprocess image, $x$;
  Num of encoder–decoder layers, $m = 4$;
  Kernel size, $k = 3$;
  Num of encoder kernels, $n^i = 64 \times 2^i$;
Num of decoder kernels, $n^j = 64 \times 2^{j-1}$;
**Output:** Predictive segmentation map, $u$;
initialization;
$$x^1 = F\left(x, k, n^i\right), (i = 0)$$
**Encoder:**
**for** $i = 1$ to $m$ do
    $z^{i+1} = \text{DTCWT\_Scat}(x^i)$
    $x^{i+1} = F(z^{i+1}, k, n \times 2^i)$
**end**
**Decoder:**
$d^{m+1} = x^{m+1}$
**for** $j = m + 1$ to $2$ do
    $p^{j-1} = \text{concate}(x^{j-1}, \text{upsample}(d^j))$
    $d^{j-1} = F(p^{j-1}, k, n \times 2^j)$
**end**
$u = \text{softmax}(conv(d^1, 1))$
**return** $u$

---

The LSW-Net framework contains a convolutional feature extraction module that is followed by batch normalization [21] after each convolution. The purpose is to accelerate the convergence speed of the LSW-Net framework and reduce the correlation between layers, see Figure 4 for details. The details can be described as follows: First, we use a $3 \times 3$ size kernel for convolution and batch normalization. Then, we use the ReLU function to activate and to achieve the purpose of nonlinear transformation. Finally, the above process is repeated once. The mathematical expression is $F(\square, k, n) = [\text{ReLU}(norm(conv(\square, k, n)))]_2$, where $\square, k, n, [\square]_2$ indicates, respectively, the input map, the size of the convolution kernel, the number of convolution kernels, and the convolution feature extraction module which is executed twice.

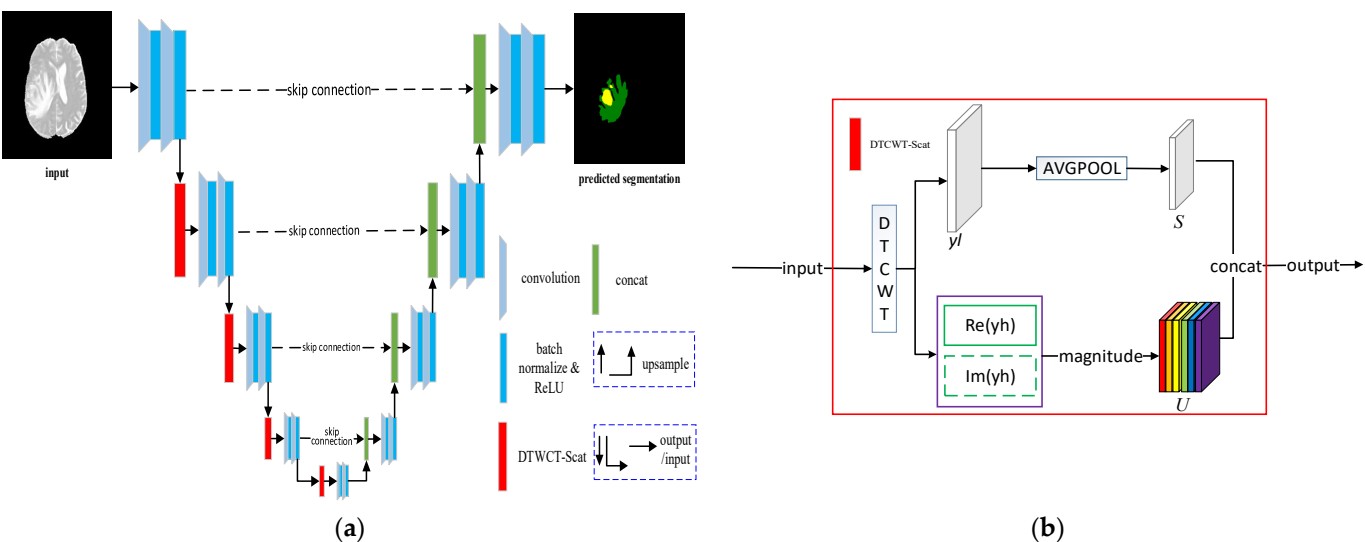

**(a)**                                   **(b)**

**Figure 4.** LSW-Net model. (**a**) LSW-Net; (**b**) DTCWT-Scat module.

### 3.2. DTCWT-Scat Module

The DTCWT-Scat module can be described in detail as follows. First, the dual-tree complex wavelet transform is performed on the feature map. Then the low-frequency information is processed using average pooling low-pass filtering, that is, $S = Avgpool(yl, 2)$, and the magnitude (*mag*) of the high-frequency real and imaginary are also calculated, i.e., $U = mag(\text{Re}(yh), \text{Im}(yh)) = \sqrt{(\text{Re}(yh))^2 + (\text{Im}(yh))^2}$. Then, we merge $S$ and $U$, as shown in Figure 4b, where $yl, yh, \text{Re}(\bullet), \text{Im}(\bullet)$ represents low-frequency information, high-frequency information, and real and imaginary operators, respectively.

The DTCWT-Scat module has two significant advantages. The first advantage is that it is able to perform a dual-tree complex wavelet transform on the input image. This transform supports the backpropagation of errors and can update the parameters so that the parameters of the previous convolution layer can be learned. Afterwards, the frequency domain features can be extracted. The second advantage is that the wavelet function has local waveform characteristics and is stable to local deformation. As a result, the LSW-Net model will be more stable and sensitive to small deformations in medical images such as tumors and will be more accurate for small feature extraction.

### 3.3. IAC-Loss Function

The flaws of CE-Loss and AC-Loss are acknowledged in Section 2.2. After absorbing the advantages of the AC-Loss function, we designed a contour segmentation minimum energy function, which can be written as follows,

$$\min_{c_1, c_2} Region \tag{1}$$

$$s.t. \int_C |\nabla u| ds = \int_C |\nabla v| ds. \tag{2}$$

where $Region = \int_\Omega ((c_1 - v)^2 - (c_2 - v)^2) \cdot u dx$, $u, v, s, C, \Omega$ represents the predicted image, segmentation image, curve arc length, segmentation contour curve, and image area, respectively. The variables $c_1, c_2$ are constant variables. Since $||\nabla u| - |\nabla v|| \leq |\nabla(u - v)|$, there is,

$$\int_C ||\nabla u| - |\nabla v|| ds \leq \int_C |\nabla(u - v)| ds. \tag{3}$$

Using the Lagrangian multiplier method, we construct a new contour segmentation energy function.

$$\min_{c_1, c_2, C} Loss_{IAC} = Region + \alpha \cdot IAC(Length), \tag{4}$$

$$IAC(Length) = \int_C |\nabla(u - v)| ds, \tag{5}$$

where $\alpha$ is the equilibrium coefficient. The first item is the area of the segmentation target area, and the second item is the difference between the target boundary length of the predicted image and the ground truth.

Figure 3 verifies the image segmentation advantages of the IAC-Loss energy function in complex backgrounds. It can be observed that when the segmentation target has no small target, $IAC(Length) = CE(Loss) = 0$, but $AC(Length) = 314$, which indicates that IAC-Loss and CE-Loss will stop during minimization, but AC-Loss will continue to decrease. When the segmentation has small targets, $IAC(Length) = 12, CE(Loss) = 1.109$, $AC(Length) = 320$, which shows that IAC-Loss not only improves the sensitivity to small boundaries but also better solves the problem of under-segmentation, see Figure 3.

In order to facilitate numerical calculation, the specific calculation discrete are also written,

$$Region = \sum_\Omega^{i=1, j=1} u_{i,j}(c_1 - v_{i,j})^2 + \sum_\Omega^{i=1, j=1} (1 - u_{i,j})(c_2 - v_{i,j})^2, \tag{6}$$

$$IAC(Length) = \sum_{\Omega}^{i=1,j=1} \left( \left| \nabla u_{x_{i,j}} - \nabla v_{x_{i,j}} \right| + \left| \nabla u_{y_{i,j}} - \nabla v_{y_{i,j}} \right| \right), \tag{7}$$

where $\alpha$ is the balance coefficient, $u_{ij} \in [0,1]$ is the predicted probability map, $v_{ij} \in \{0,1\}$ is the binary code of the ground truth, and $c_1, c_2$ can be defined as a constant of 1 or 0. $\nabla u_{x_{i,j}}, \nabla u_{y_{i,j}}, \nabla v_{x_{i,j}}, \nabla v_{y_{i,j}}$ are the differences of $u_{i,j}$ and $v_{i,j}$ in the horizontal and vertical directions, respectively. The algorithm flow is shown in Algorithm 2.

---

**Algorithm 2:** Improved AC-Loss function

---

**Input:** Predictive segmentation map, $u$;
       Binary ground truth map, $v$;
       Equilibrium coefficient, $\alpha$;
       Batch size, $B$; Channels, $C$;
       Image width, $W$; Image height, $H$;
**Output:** IAC-Loss Error, $Loss_{IAC}$;
initialization;
$c_{in} = [1]_{B \times C \times W \times H}, c_{out} = [0]_{B \times C \times W \times H}$
$Region_{in} = u \times (v - c_{in})^2$
$Region_{out} = (1 - u) \times (v - c_{out})^2$
$Region = Region_{in} + Region_{out}$
$\nabla h_x = h[:,:, 1 :,:] - h[:,:,:-1,:], h = u, v$
$\nabla h_y = h[:,:,:, 1 :] - h[:,:,:,:-1], h = u, v$
$IAC(Length) = \left| \nabla u_x - \nabla v_x \right| + \left| \nabla u_y - \nabla v_y \right|$
$Loss_{IAC} = Region + \alpha \cdot IAC(Length)$
**return** $Loss_{IAC}$

---

## 4. Experiments

In the following experiments, we use the DRIVE [22] and MICCAI-BraTS2020 [23] datasets. In the experimental results, the BraTS brain tumor segmentation evaluation metrics were recorded when $epoch = 200$, and the DRIVE retinal blood vessel segmentation evaluation metrics were recorded when $epoch = 10$.

All models are trained on an i7-10750H, NVIDIA RTX 2070 GPU with 8G RAM. The Python language is used for programming and the deep learning framework used is Pytorch.

### 4.1. Data Preprocessing and Evaluation Metrics

The BraTS2020 brain tumor dataset has 369 patient samples, and each patient contains 4 modalities of MRI image data. After splicing and slicing the four-modal data, slices are obtained. In this experiment, 297 samples are randomly selected as the training set and validation set, and the remaining 72 samples are reserved as the test set. After removing the slices without lesions there are still 19,874 slices, of which 80% of the slices are randomly selected as the training set and 20% of the slices are selected as the validation set. In the DRIVE retinal dataset, the first 20 images are selected as the training set and validation set and the last 20 images are used as the test set.

In the DRIVE retinal dataset, the first 20 images are selected as the training set and validation set, and the last 20 images are used as the test set. In this experiment, since there are only a small number of sample sets in the DRIVE retinal dataset, we preprocess the images of the training set according to image rotation, horizontal flip, vertical flip, translated, and random cropping, in order to expand the sample size of the training set.

In this paper, our model quality is evaluated in terms of the standard evaluation metrics such as precision, Dice coefficient, sensitivity, specificity, and accuracy, which are shown in Table 1. $TP, FP, FN, TN$ represent true positives, false positives, false negatives, and true negatives, respectively.

**Table 1.** Evaluation metrics.

| Metric | Description |
|---|---|
| **Pre** (Precision) | $\frac{TP}{FP+TP}$ |
| **Dice** (Dice coefficient) | $\frac{2\cdot TP}{2\cdot TP+FP+FN}$ |
| **Sen** (Sensitivity) | $\frac{TP}{TP+FN}$ |
| **Spe** (Specificity) | $\frac{TN}{FP+TN}$ |
| **Acc** (Accuracy) | $\frac{TP+TN}{FP+FN+TP+TN}$ |

*4.2. Experiment 1: BraTS Brain Tumor Segmentation*

In this subsection, we will compare the evaluation metrics of the LSW-Net model with FCN, SegNet, and At-Unet models, as well as the 3D visualization. We use a combination of binary cross entropy (BCE) and Dice loss to train the LSW-Net. The loss is formulated as:

$$loss_{BraTs} = loss_{Dice} + 0.5 \cdot loss_{BCE}, \tag{8}$$

where $loss_{Dice} = 1 - \frac{2\sum y_i \cdot \hat{y}_i}{\sum y_i + \sum \hat{y}_i}$, $loss_{BCE} = -\frac{1}{N}\sum[y_i \cdot \log(\hat{y}_i) + (1-y_i) \cdot \log(\hat{y}_i)]$, $y_i \in \{0,1\}$ is the binary-coded value of the ground truth, and $\hat{y}_i \in [0,1]$ is the predicted value.

In Table 2, the evaluation metrics of the LSW-Net model are recorded, where ET, TC, WT, and AVG represent the enhanced tumor area, tumor core, the entire tumor area, and the average metric, respectively. After comparison to the classic FCN, SegNet, and At-Unet models, it can be observed that the Dice coefficient, accuracy, and sensitivity of the LSW-Net model are all excellent. The LSW-Net model improved the Dice coefficient, accuracy, and sensitivity by at least 3.51%, 2.11%, and 0.46%, respectively.

**Table 2.** Comparison of LSW-Net model with classical segmentation algorithms on BraTS2020.

| Method | Pre | | | | Dice | | | | Sen | | | |
|---|---|---|---|---|---|---|---|---|---|---|---|---|
| | ET | TC | WT | AVG | ET | TC | WT | AVG | ET | TC | WT | AVG |
| FCN [7] | 0.7650 | 0.6554 | 0.7831 | 0.7345 | 0.7656 | 0.6802 | 0.8125 | 0.7528 | 0.8197 | 0.7904 | 0.8722 | 0.8274 |
| SegNet [11] | 0.7748 | 0.7076 | 0.8669 | 0.7831 | 0.7316 | 0.6984 | 0.8448 | 0.7583 | 0.7615 | 0.7754 | 0.8464 | 0.7944 |
| At-Unet [12] | 0.7764 | 0.7235 | 0.8791 | 0.7930 | 0.7646 | 0.7312 | 0.8600 | 0.7853 | 0.8080 | 0.8240 | 0.8665 | 0.8328 |
| **LSW-Net (Ours)** | **0.8319** | **0.7447** | **0.9077** | **0.8281** | **0.7947** | **0.7448** | **0.8797** | **0.8064** | **0.8125** | **0.8308** | **0.8690** | **0.8374** |

Figure 5 shows the 2D visualization comparison of the segmentation results of four brain tumor samples between the LSW-Net model and the classic FCN, SegNet, At-Unet models. After comparison with the ground truth, it can be observed that the LSW-Net model performs better than the three classical models in terms of segmentation and is more suitable for BraTS brain tumor dataset image segmentation. In the segmentation results in line one of Figure 5, it can be observed that the other three classic models have misclassified in the enhanced tumor area and the edema area. Conversely, the LSW-Net model has a clear and complete outline, which also shows the validity of the LSW-Net model.

The LSW-Net model segmentation results have fewer outliers and mis-segmented blocks, so they are closer to ground truth when compared to the 3D visualization of the classical FCN, SegNet, and At-Unet segmentation results. The 3D visualization in Figure 6 shows that the LSW-Net model has achieved a good overall segmentation effect. This contributes to a clearer understanding and judgment of tumor size, boundary, and location.

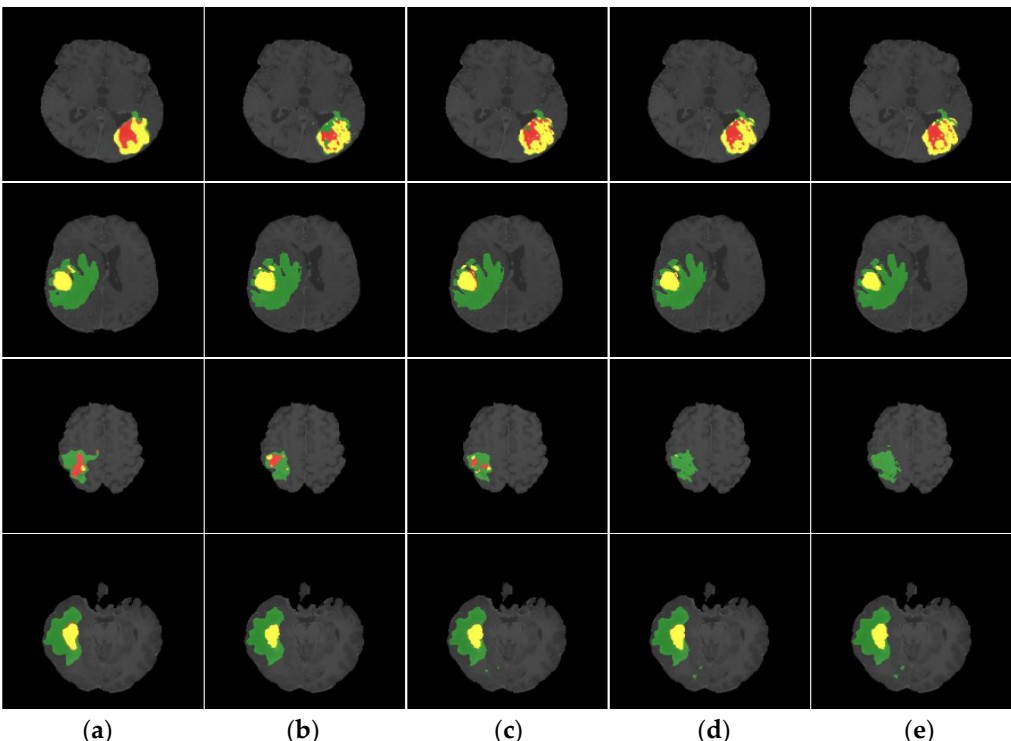

**Figure 5.** Comparison of segmentation results of four brain tumor samples, in which red, yellow, and green indicate necrotic area, enhanced tumor area, and edema area, respectively. (**a**) FCN; (**b**) SegNet; (**c**) At-Unet; (**d**) LSW-Net (Ours); (**e**) Ground truth.

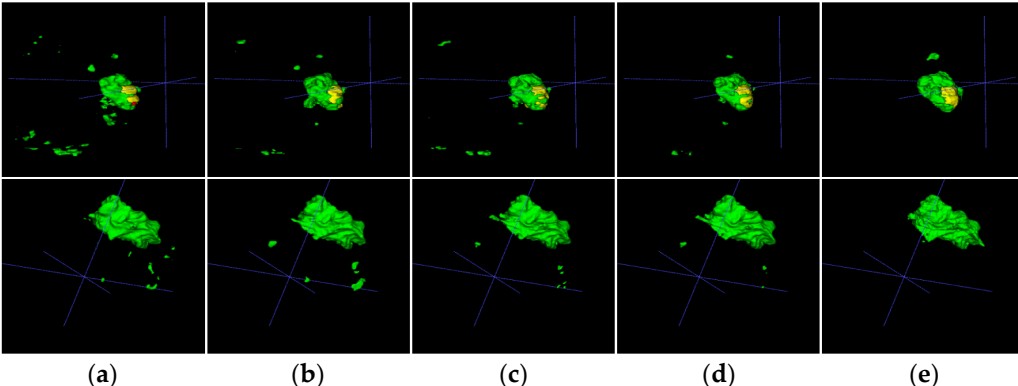

**Figure 6.** 3D visualization comparison of the segmentation results of two brain tumors, in which red, yellow, and green indicate the necrotic area, the enhanced tumor area, and the edema area, respectively. (**a**) FCN; (**b**) SegNet; (**c**) At-Unet; (**d**) LSW-Net (Ours); (**e**) Ground truth.

In addition to the experimental comparison of the LSW-Net with the classical segmentation algorithms, the performance of the LSW-Net model, using the BraTS brain tumor dataset, is assessed here against several advanced segmentation algorithms developed in recent years by researchers such as Zhang et al. [24], Li et al. [25], Feng et al. [26], Latif et al. [27] and Hao et al. [28], see Table 3. The comparison assesses performance in terms of the Dice coefficients for ET, TC, WT, and average (AVG). The evaluation metrics in Table 3 show that the LSW-Net model and these advanced segmentation algorithms have advantages and disadvantages in ET, TC, and WT indicators. However, the most important evaluation indicator is the average indicator of the Dice coefficient, that is, the AVG indicator. The AVG index of the LSW-Net model is the highest, which also shows that our model has better segmentation performance on the BraTS2020 dataset.



**Table 3.** Comparison of LSW-Net model with some advanced algorithms on BraTS2020.

| Method | Year | Dice | | | |
|---|---|---|---|---|---|
| | | ET | TC | WT | AVG |
| Zhang et al. [24] | 2019 | 0.7070 | 0.7380 | 0.8850 | 0.7767 |
| Li et al. [25] | 2019 | 0.7450 | **0.8080** | 0.8650 | 0.8060 |
| Feng et al. [26] | 2020 | 0.7100 | 0.7300 | **0.9000** | 0.7800 |
| Latif et al. [27] | 2021 | 0.7180 | 0.7460 | 0.8960 | 0.7860 |
| Hao et al. [28] | 2021 | 0.7926 | 0.7465 | 0.8764 | 0.8051 |
| **LSW-Net (Ours)** | | **0.7947** | 0.7448 | 0.8797 | **0.8064** |

*4.3. Experiment 2: DRIVE Retinal Segmentation*

In this subsection, we will verify the effectiveness of the IAC-Loss function for segmentation on the DRIVE retina dataset. In the experiment, the LSW-Net model will be used as the backbone network and the loss function will be the IAC-Loss function, denoted as the LSW-Net + IAC-Loss model. Finally, the LSW-Net + IAC-Loss model segmentation results are compared with other 14 models, including Cheng et al. [29], Azzopardi et al. [30], Roychowdhury et al. [31], DRIU [32], HED [33], Unet [34], Recurrent Unet [34], R2Unet [34], Guo et al. [35], Du et al. [36], Arias et al. [37], Zou et al. [38], and MD-Net [39] models. In addition, two examples of segmentation effects are shown in terms of overall and local details. Compared with the other 14 models in Table 4, it can be seen that the LSW-Net + IAC-Loss model is higher than the other 14 algorithms in terms of Dice coefficient and specificity; it is second only to the MD-Net [39] model in the accuracy index. Compared with the segmentation results of the classic Unet model, the sensitivity of the LSW-Net model offers an improvement of 3.39%, which is an obvious improvement and has a significant effect. These advantages indicate that our model performs well.

**Table 4.** Comparison of LSW-Net + IAC-Loss model with some advanced models on DRIVE.

| Method | Year | Dice | Sen | Spe | Acc |
|---|---|---|---|---|---|
| Cheng et al. [29] | 2014 | - | 0.7252 | 0.9798 | 0.9474 |
| Azzopardi et al. [30] | 2015 | - | 0.7655 | 0.9704 | 0.9442 |
| Roychowdhury et al. [31] | 2016 | - | 0.7250 | 0.9830 | 0.9520 |
| DRIU [32] | 2016 | 0.6701 | **0.9696** | 0.9115 | 0.9165 |
| HED [33] | 2017 | 0.6400 | 0.9563 | 0.9007 | 0.9054 |
| Unet [34] | 2019 | 0.8142 | 0.7537 | 0.9820 | 0.9553 |
| Recurrent Unet [34] | 2019 | 0.8155 | 0.7751 | 0.9816 | 0.9556 |
| R2Unet [34] | 2019 | 0.8171 | 0.7792 | 0.9813 | 0.9556 |
| Guo et al. [35] | 2020 | 0.8215 | 0.8283 | 0.9726 | 0.9542 |
| Du et al. [36] | 2021 | - | 0.7814 | 0.9810 | 0.9556 |
| Arias et al. [37] | 2021 | - | 0.8597 | 0.9690 | 0.9563 |
| Zou et al. [38] | 2021 | 0.8129 | 0.7761 | 0.9792 | 0.9519 |
| MD-Net [39] | 2021 | 0.8099 | 0.8065 | 0.9826 | **0.9676** |
| MFE-Net [40] | 2022 | 0.8204 | 0.7853 | 0.9812 | 0.9563 |
| **LSW-Net + IAC-Loss (Ours)** | | **0.8216** | 0.7876 | **0.9837** | 0.9565 |

The comparison experiment, using the segmentation results from the DRIVE retinal blood vessel dataset, is shown in Figure 7. In the first and third rows of Figure 7 it can be observed that the segmentation results of DRIU [32] and HED [33] have obvious oversegmentation. In lines two and four of Figure 7, it can be seen that the segmentation results of the LSW-Net + IAC-Loss model have less noise and clearer contours. Through this experimental comparison, it can be shown that the LSW-Net + IAC-Loss model has better segmentation effectiveness.

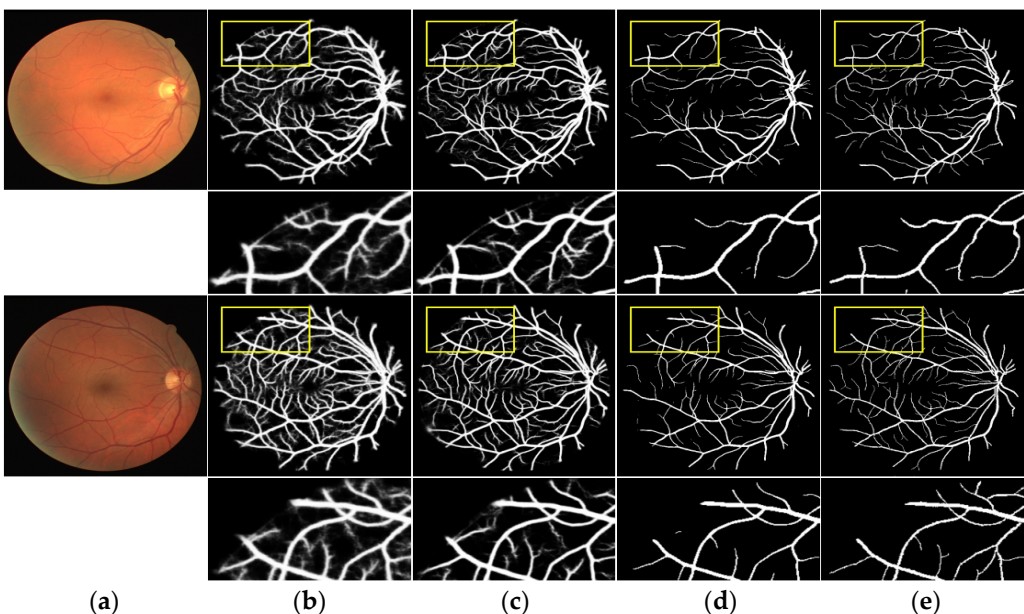

(a) (b) (c) (d) (e)

**Figure 7.** Comparison of LSW-Net + IAC-Loss with advanced segmentation models results on DRIVE. Lines 2 and 4 are the local details. (**a**) Test Image; (**b**) HED; (**c**) DRIU; (**d**) LSW-Net + IAC-Loss (Ours); (**e**) Ground truth.

*4.4. Experiment 3: Discussion on Equilibrium Coefficient $\alpha$*

To evaluate the effect of the balance coefficient $\alpha$ in the LSW-Net + IAC-Loss model, ablation experiments are performed on the DRIVE dataset in this subsection.

In the experiment, the evaluation metrics of the LSW-Net + IAC-Loss model are recorded in Table 5, where $\alpha \in [0.1, 0.5]$. Table 5 shows that the differences in each metric are not obvious; however, they all reach a high level, which also shows that the LSW-Net + IAC-Loss model has better robustness to the balance coefficient $\alpha$. When $\alpha = 0.3$, the specificity is the highest while the Dice coefficient and sensitivity are relatively low. Alternatively, when $\alpha = 0.1$ or $\alpha = 0.5$, the Dice coefficient and sensitivity index values increase. Therefore, we suggest that the metrics can be fine-tuned by controlling the balance coefficient $\alpha$ according to actual needs.

**Table 5.** Influence of $\alpha$ balance coefficient on LSW-Net + IAC-Loss model segmentation result indicators.

| $\alpha$ | Pre | Dice | Sen | Spe | Acc |
|---|---|---|---|---|---|
| 0.1 | 0.8525 | **0.8231** | 0.9565 | **0.7957** | 0.9799 |
| 0.2 | 0.8542 | 0.8222 | 0.9564 | 0.7925 | 0.9802 |
| 0.3 | 0.8588 | 0.8216 | 0.9565 | 0.7876 | **0.9837** |
| 0.4 | **0.8602** | 0.8221 | **0.9566** | 0.7873 | 0.9813 |
| 0.5 | 0.8571 | 0.8227 | **0.9566** | 0.7909 | 0.9807 |

*4.5. Experiment 4: IAC-Loss Effectiveness Evaluation*

In this subsection, we further evaluate the advantages of the IAC-Loss function. For the segmentation of the DRIVE retina dataset, the LSW-Net is used as the backbone network and the loss functions are the CE-Loss, AC-Loss, and IAC-Loss functions, denoted as +CE-Loss, +AC-Loss, and +IAC-Loss models, respectively. The segmentation results are shown in Table 6.

**Table 6.** Comparison of IAC-Loss with related loss function metrics.

|  | Dice | Sen | Spe | Acc |
|---|---|---|---|---|
| +AC-Loss | 0.7875 | 0.7147 | **0.9853** | 0.9509 |
| +CE-Loss | 0.8182 | 0.7920 | 0.9789 | 0.9551 |
| **+IAC-Loss ($\alpha$ = 0.1)** | **0.8231** | **0.7957** | 0.9799 | **0.9565** |

Compared with the +AC-Loss model, the +IAC-Loss model improves on the Dice coefficient and sensitivity by 3.56% and 8.1%, respectively. The accuracy is also increased by 0.56%. This illustrates the effectiveness of the IAC-Loss function for image segmentation, see Table 6.

After comparing the enlarged details of lines two and four in Figure 8, it can be seen that the boundary contour segmentation of the +IAC-Loss model is the best. Through the comparative experiments above, it can be determined that the IAC-Loss function has greater advantages for complex image boundary contours.

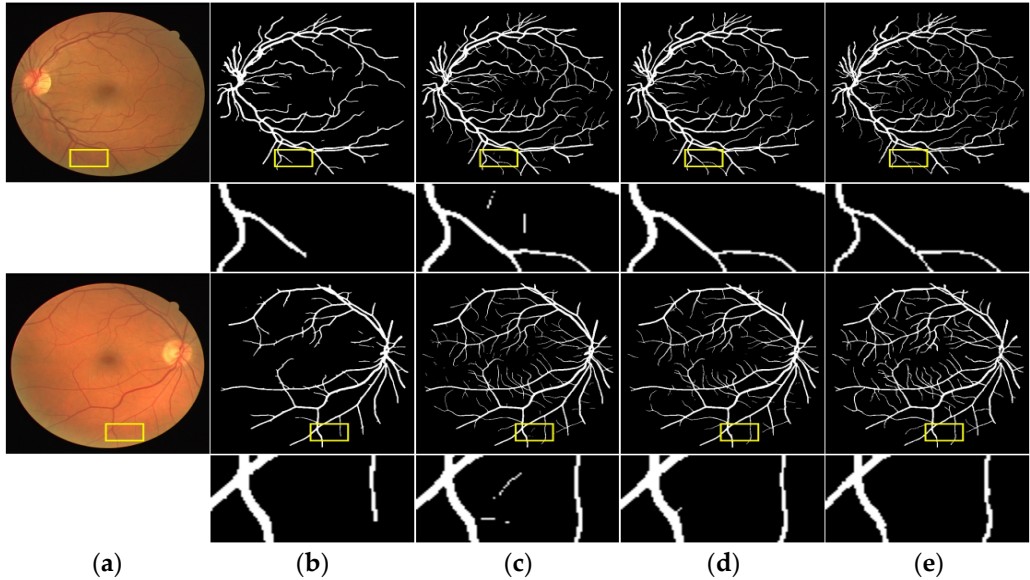

(**a**)  (**b**)  (**c**)  (**d**)  (**e**)

**Figure 8.** Comparison of +IAC-Loss with related loss function results. Lines 2 and 4 are the local details respectively. (**a**) Test Image; (**b**) +AC-Loss; (**c**) +CE-Loss; (**d**) +IAC-Loss; (**e**) Ground truth.

## 5. Conclusions

In this research, we have proposed the LSW-Net model for the BraTS2020 dataset, which achieved good experimental simulation results on the segmentation discontinuity problem. We have constructed an LSW-Net + IAC-Loss model in order to solve the weak boundary problem of small blood vessels in the DRIVE retinal vessel dataset. After introducing the dual-tree complex wavelet transform, the experimental results show that the LSW-Net has the ability to extract features and achieve better segmentation results. In the future we will further integrate the attention mechanism and the transformer method to design a better image segmentation network model.

**Author Contributions:** Methodology, R.L.; software, H.N.; validation, H.N. and Y.Z.; formal analysis, R.L.; resources, R.L., T.X. and Z.Y.; project administration, R.L.; funding acquisition, R.L. and Z.Y.; data curation, T.X. and Z.Y.; writing—review and editing, R.L.; visualization, H.N. All authors have read and agreed to the published version of the manuscript.

**Funding:** This work was supported in part by: Chongqing Natural Science Foundation under Grants No. cstc2019jcyj-msxmX0500 and No. cstc2019jcyj-msxmX0240; Technology Research Program of Chongqing Education Commission under Grand No. KJQN202001129.

**Data Availability Statement:** The DRIVE and BraTS2020 datasets are publicly available [22,23], https://drive.grand-challenge.org/. https://www.med.upenn.edu/cbica/brats2020/.

**Conflicts of Interest:** The authors declare no conflict of interest.

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
