# Peer review of "LSW-Net: A Learning Scattering Wavelet Network for Brain Tumor and Retinal Image Segmentation"

_electronics, doi:10.3390/electronics11162616_

Round 1
Reviewer 1 Report
The paper is written and organized well. Algorithms have clear description of input and output. Moreover, the experimentation is perform in adequate manner and comparsions of the scheme are made with several state of the art studies. I give the paper an accept from my side.
Author Response
请参阅附件。

Reviewer 2 Report
The proposed paper tackles the issue of medical image segmentation by proposing two main contributions. The first is the design of a Dual-Tree Complex Wavelet Scattering Transform (DTCWT-Scat) module that can be used to modify traditional deep learning architectures for segmentation like U-Net. The second is the proposal of an Improved Active Contour (IAC) loss function that attempts to offset the flaws of the existing losses (e.g. cross-entropy or active contour losses). Experiments carried out on two image datasets (one for brain tumor, the other for retina) show the superiority of a U-Net architecture modified with DTCWT-Scat over other state-of-the-art segmentation models, and the benefits of the IAC Loss over the traditional loss functions.
The methods presented in the paper are scientifically sound and look very interesting. The justification for such new methods is also well explained in the introduction. Very extensive experiments comparing the proposed approach to the state-of-the-art were carried out by the authors.
Regarding points that could be improved, I would first have two comments on the experimental aspect of the paper:
1- Have the authors attempted to evaluate the LSW-Net with the IAC-Loss on the BraTS dataset? If not, I would suggest the authors to indicate the reason why.
2- In experiment 3 described in section 4.4., why was the parameter alpha constrained to [0.1,0.5]? Couldn't larger values for alpha have been tested too?
My other remarks concern more minor details, and are nearly exclusively related to improving how the proposed method is described in the manuscript:
3- In the caption of Figure 1, please indicate what the yellow frames represent.
4- In the description of algorithm 1, I would advise to use letters instead of numerical values (e.g. 3->k, 64 -> n), and introduce all notations (e.g. F, k, n) in the algorithm description (even if they are described at a later point in the manuscript).
5- There seems to be a contradiction/typo in section 3.2: the authors mention that the magnitude component in the DTCWT-Scat module is computed from the real and imaginary parts of the high-frequency component, but the equation at line 191 uses Im(yl) instead of Im(yh).
6- In section 3.3, it would have been good if the authors could indicate what are the main differences (definition-wise) between their proposed loss and the AC-Loss from the literature.
7- In equations (1), (2), (3) and (5), I would suggest the authors to indicate which terms under the integrals depend on the integration variables x or s.
8- In the description of algorithm 2, please introduce the variables B, C, W and H.
9- Although the paper can be fairly easily understood, I would recommend the authors to perform an additional English proofreading check to correct some awkwardly formulated sentences that can be found at some points in the manuscript.
Reviewer 3 Report
Figure 4, typo DTWCT / DTCWT. Also missing legend for the green boxes in (a).
Page 6, section 3.3, what is the reasoning for the IAC-loss energy function to improve under-segmentation compared with AC-loss energy function?
Page 10, line 343, what is ablation experiment to characterize equilibrium coefficient?
Reviewer 4 Report
The authors proposed a Learning Scattering Wavelet Network for BrainTumor and Retinal Image Segmentation. The work is interesting and I have few minor suggestions.
1. Don't cite references in a bunch like [1-5] etc. all the refs should be discussed in detail.
2. How the DTCWT-Scat help to separate the high-frequency and low-frequency feature is not very clear.
3. literature review section is poor. add more recent works like "https://doi.org/10.1007/978-981-16-6887-6_11"
"https://doi.org/10.1155/2022/9505229"
4. The parameters used in the proposed algorithm are not discussed in detail. Add simulation parameters details in a table.
5. Comparison with more recent works is more appreciated.
6. English language editing is required, make sure all the refs, eqns, and figs are cited in the text.
Round 2
Reviewer 2 Report
The authors have answered my requests in a satisfying way. I therefore recommend the acceptance of this article.